# A Fast Convoluted Story:
# Scaling Probabilistic Inference for Integer Arithmetic

**Lennert De Smet**
KU Leuven
Belgium

**Pedro Zuidberg Dos Martires**
Örebro University
Sweden

## Abstract

As illustrated by the success of integer linear programming, linear integer arithmetic is a powerful tool for modelling combinatorial problems. Furthermore, the probabilistic extension of linear programming has been used to formulate problems in neurosymbolic AI. However, two key problems persist that prevent the adoption of neurosymbolic techniques beyond toy problems. First, probabilistic inference is inherently hard, #P-hard to be precise. Second, the discrete nature of integers renders the construction of meaningful gradients challenging, which is problematic for learning. In order to mitigate these issues, we formulate linear arithmetic over integer-valued random variables as tensor manipulations that can be implemented in a straightforward fashion using modern deep learning libraries. At the core of our formulation lies the observation that the addition of two integer-valued random variables can be performed by adapting the fast Fourier transform to probabilities in the log-domain. By relying on tensor operations we obtain a differentiable data structure, which unlocks, virtually for free, gradient-based learning. In our experimental validation we show that tensorising probabilistic linear integer arithmetic and leveraging the fast Fourier transform allows us to push the state of the art by several orders of magnitude in terms of inference and learning times.

## 1 Introduction

Integer linear programming (ILP) [15, 32] uses linear arithmetic over integer variables to model intricate combinatorial problems and has successfully been applied to domains such as scheduling [31], telecommunications [34] and energy grid optimisation [26]. If one replaces deterministic integers with integer-valued random variables, the resulting probabilistic arithmetic expressions can be used to model probabilistic combinatorial problems. In particular, many problems studied in the field of neurosymbolic AI can be described using probabilistic linear integer arithmetic.

Unfortunately, exact probabilistic inference for integer arithmetic is a #P-hard problem in general. Consequently, even state-of-the-art probabilistic programming languages with dedicated inference algorithms for discrete random variables, such as ProbLog [8] and Dice [13], fail to scale. The reason being that they resort to exact enumeration algorithms, as exemplified in Figure 1. Note that while approximate inference algorithms such as Monte Carlo methods and variational inference can be applied to probabilistic combinatorial problems, they come with their own set of limitations, as discussed by Cao et al. [5]. For instance, conditional inference with low-probability evidence.

In order to mitigate the computational hardness of probabilistic inference over integer-valued random variables, we make the simple yet powerful observation that the probability mass function (PMF) of the sum of two random variables is equal to the convolution of the PMFs of the summands. The key advantage of this perspective is that the exact convolution for finite domains can be implemented efficiently using the fast Fourier transform (FFT) in $\mathcal{O}(N \log N)$, which avoids the traditionally quadratic behaviour of computing the PMF of a sum of two random variables (Figure 1). Moreover,

38th Conference on Neural Information Processing Systems (NeurIPS 2024).

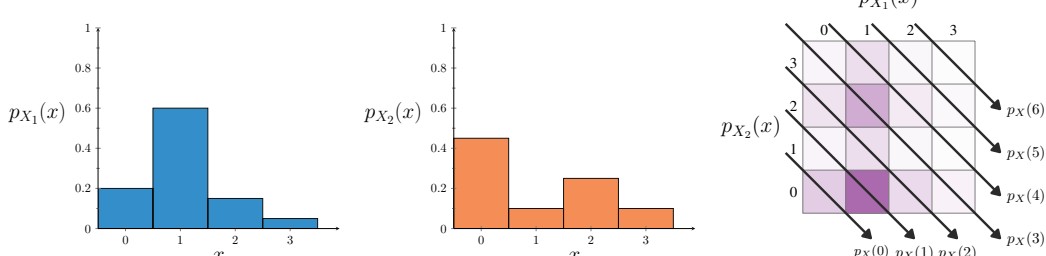

Figure 1: On the left and in the middle we have two histograms representing the probability distributions of the random variables $X_1$ and $X_2$, respectively. The grid on the right represents the joint probability of the two distributions, with more intense colors indicating events with higher probability. The distribution of the random variable $X = X_1 + X_2$ can be obtained by summing up the diagonals of the grid as indicated in the figure. While this method of obtaining the distribution for $X$ is valid and used by state-of-the-art neurosymbolic techniques [14, 21], the explicit construction of the joint is unnecessary and hampers inference and learning times (cf. Section 5).

efficient implementations of the FFT are readily available in modern deep learning libraries such as TensorFlow [1] and PyTorch [28], making our approach to probabilistic inference end-to-end differentiable by construction. In turn, differentiability allows us to apply our approach to prototypical problems in neurosymbolic AI.

Our main contributions are the following. **1)** We propose a tensor representation of the distributions of bounded integer-valued random variables that allows for the computation of the distribution of a sum of two such variables in $\mathcal{O}(N \log N)$ instead of $\mathcal{O}(N^2)$ by exploiting the fast Fourier transform (Section 2). **2)** We formulate common operations in linear integer arithmetic, such as multiplication by constants and the modulo operation, as tensor manipulations (Section 3). These tensorised operations give rise to PLIA$_t$, a scalable and differentiable framework for Probabilistic Linear Integer Arithmetic.[1] PLIA$_t$ supports two exact probabilistic inference primitives; taking expected values and performing probabilistic branching (Section 4). **3)** We provide experimental evidence that PLIA$_t$ outperforms the state of the art in exact probabilistic inference for integer arithmetic [5] in terms of inference time by multiple orders of magnitude (Section 5.1). Moreover, we deploy PLIA$_t$ in the context of challenging neurosymbolic combinatorial problems, where it is again orders of magnitude more efficient when compared to state-of-the-art *exact and approximate* methods (Section 5.2).

## 2 Efficient Addition of Integer-Valued Random Variables

In what follows, we denote random variables by uppercase letters, while a specific realisation of a random variable is written in lowercase. That is, the value $x$ is an element of the sample space $\Omega(X)$ of $X$. We will also refer to $\Omega(X)$ as the domain of the random variable X. Furthermore, $\Omega(X)$ is assumed to be integer-valued, i.e. it is a finite subset of the integers $\mathbb{Z}$ with lower and upper bounds $L(X)$ and $U(X)$, respectively. In particular, we have that the cardinality $|\Omega(X)| = U(X) - L(X) + 1$. We will call these integer-valued random variables *probabilistic integers* from here on.

The distribution of a probabilistic integer $X$ is represented using its probability mass function (PMF) $p_X : \Omega(X) \to [0, 1]$ with the conventional restrictions

$$\forall x \in \Omega(X) : p_X(x) \geq 0 \qquad \text{and} \qquad \sum_{x \in \Omega(X)} p_X(x) = 1. \qquad (1)$$

### 2.1 Probabilistic Integers and the Convolution Theorem

At the core of PLIA$_t$ and linear arithmetic in general is the addition of two probabilistic integers $X_1$ and $X_2$. Let us assume for now that $X_1$ and $X_2$ satisfy $L(X_1) = L(X_2) = 0$ and have upper bounds $U(X_1) = N_1$ and $U(X_2) = N_2$. Just as in Figure 1, we would now like to find the PMF of the random variable $X$ such that $X = X_1 + X_2$. However, contrary to Figure 1, we wish to avoid the

---

[1]The subscript "t" in PLIA$_t$ stands for "tensorised" and is not pronounced.

explicit quadratic construction of all possible outcomes. To this end, we exploit that the PMF of the sum of two random variables is equal to the convolution of their respective PMFs [12]

$$p_X(x) = (p_{X_1} * p_{X_2})(x), \qquad X = X_1 + X_2. \tag{2}$$

Next, we apply the Fourier transform $\mathcal{F}$ to both sides of the equation and use the convolution theorem (CT) [23] that states that the Fourier transform of two convoluted functions is equal to the product of their transforms

$$\mathcal{F}(p_X)(x) = \mathcal{F}(p_{X_1} * p_{X_2})(x) \overset{\text{CT}}{=} \mathcal{F}(p_{X_1})(x) \cdot \mathcal{F}(p_{X_2})(x) = \widehat{p}_{X_1}(x) \cdot \widehat{p}_{X_2}(x), \tag{3}$$

where we also introduce the hat notation $\mathcal{F}(p_X) = \widehat{p}_X$ for a Fourier transformed PMF $p_X$. As $X = X_1 + X_2$, we know that $\Omega(X) = \{0, \ldots, N_1 + N_2\}$. Consequently, the PMF $p_X$ is non-zero for just these $M = N_1 + N_2 + 1$ domain elements and can be represented using a vector of probabilities

$$\boldsymbol{\pi}_X[x] = p_X(x), \quad \forall x \in \{0, \ldots, N_1 + N_2\}. \tag{4}$$

All vectors of probabilities will be written using a boldface $\boldsymbol{\pi}$ and their elements accessed using square brackets. Looking at Equation 3, we would now like to express the Fourier transformed probability vector $\boldsymbol{\pi}_X$ as the point-wise product of the transformed vectors

$$F_M \boldsymbol{\pi}_X = \widehat{\boldsymbol{\pi}}_{X_1} \odot \widehat{\boldsymbol{\pi}}_{X_2}, \tag{5}$$

where $F_M \in \mathbb{C}^{M \times M}$ is the $M$-point discrete Fourier transform (DFT) matrix and the symbol $\odot$ denotes the Hadamard product. In order for Equation 5 to hold we need to have that

$$\widehat{\boldsymbol{\pi}}_{X_1} = F_M \boldsymbol{\pi}_{X_1} \qquad \text{and} \qquad \widehat{\boldsymbol{\pi}}_{X_2} = F_M \boldsymbol{\pi}_{X_2}. \tag{6}$$

At first sight these equalities seem to cause a problem; each probabilistic integer $X_i$ has a domain of size $N_i + 1$ while its PMF should be represented with a probability vector $\boldsymbol{\pi}_{X_1} \in \mathbb{R}^M$ for the multiplication with $F_M$ to make sense. Fortunately, this problem is easily resolved by observing that we can extend the domain $\Omega(X_i)$ of $X_i$ by simply assigning a probability of zero to newly added elements. In practice, we simply pad the probability vectors $\boldsymbol{\pi}_{X_1}$ and $\boldsymbol{\pi}_{X_2}$ with $N_2$ and $N_1$ zeros at the end to obtain vectors of dimension $M$. With this issue resolved, we can finally obtain the probability vector $\boldsymbol{\pi}_X$ that represents the PMF $p_X$ by using Equation 5 via

$$F_M \boldsymbol{\pi}_X = \widehat{\boldsymbol{\pi}}_{X_1} \odot \widehat{\boldsymbol{\pi}}_{X_2} \Leftrightarrow F_M \boldsymbol{\pi}_X = F_M \boldsymbol{\pi}_{X_1} \odot F_M \boldsymbol{\pi}_{X_2} \tag{7}$$

$$\Leftrightarrow \boldsymbol{\pi}_X = F_M^{-1} \left( F_M \boldsymbol{\pi}_{X_1} \odot F_M \boldsymbol{\pi}_{X_2} \right). \tag{8}$$

## 2.2 The Fast Log-Conv-Exp Trick

The attentive reader might have noticed that, even though we avoid the explicit construction of the joint probability distribution $p_{X_1 X_2}(x_1, x_2)$, we have not gained much. The matrix-vector products in Equation 8 still take $\mathcal{O}(M^2)$ to compute. Fortunately, matrix-vector products where the matrix is the DFT matrix or its inverse can be computed in time $\mathcal{O}(M \log M)$ by using the fast Fourier transform (FFT), with $M$ being the size of the vector. As a result, we can express Equation 8 as

$$\boldsymbol{\pi}_X = \text{IFFT} \left( \text{FFT}(\boldsymbol{\pi}_{X_1}) \odot \text{FFT}(\boldsymbol{\pi}_{X_2}) \right). \tag{9}$$

Computing the values of the vector $\boldsymbol{\pi}_X$ can now be done in time $\mathcal{O}(M \log M)$. First we apply the FFT on the vectors $\boldsymbol{\pi}_{X_1}$ and $\boldsymbol{\pi}_{X_2}$. Then we multiply the transformed vectors pointwise and apply the inverse FFT on the result of this Hadamard product. We note that Equation 9 is a well known result from the signal processing literature, where convolutions are always computed in this fashion [23].

However, applying Equation 9 naively to the problem of probabilistic inference quickly results in numerical stability issues. The problem is that multiplying together small probabilities eventually results in numerical underflow. A well-known and widely used remedy to this problem is the log-sum-exp trick, which allows one to avoid underflow by performing computations in the log-domain instead of the linear domain. Inspired by the log-einsum-exp trick [29], we introduce the fast log-conv-exp trick, which allows us to perform the FFT on probabilities in the log-domain.

We first characterise a probability distribution $p_X$ not by the vector of probabilities $\boldsymbol{\pi}_X$ but by the vector of log-probabilities $\boldsymbol{\lambda}_X = \log \boldsymbol{\pi}_X$. In terms of log-probabilities, Equation 9 can be written as

$$\exp \boldsymbol{\lambda}_X = \text{IFFT} \left( \text{FFT}(\exp \boldsymbol{\lambda}_{X_1}) \odot \text{FFT}(\exp \boldsymbol{\lambda}_{X_2}) \right). \tag{10}$$

Now define $\mu_i := \max_{x \in \Omega(X_i)} \boldsymbol{\lambda}_{X_i}[x]$ as the maximum value present in the vector $\boldsymbol{\lambda}_{X_i}$, which lets us write Equation 10 as

$$\exp \boldsymbol{\lambda}_X = \text{IFFT}\Big(\text{FFT}(\exp(\boldsymbol{\lambda}_{X_1} - \mu_{X_1} + \mu_{X_1})) \odot \text{FFT}(\exp(\boldsymbol{\lambda}_{X_2} - \mu_{X_2} + \mu_{X_2}))\Big) \quad (11)$$

$$= \text{IFFT}\Big(\text{FFT}(\exp(\boldsymbol{\lambda}_{X_1} - \mu_{X_1})) \odot \text{FFT}(\exp(\boldsymbol{\lambda}_{X_2} - \mu_{X_2}))\Big) \exp(\mu_{X_1}) \exp(\mu_{X_2}).$$

Crucially, we were able to pull out the scalars $\exp(\mu_{X_1})$ and $\exp(\mu_{X_2})$ due to the linearity of the FFT transform and its inverse. Taking the logarithm of both sides results in the fast log-conv-exp trick

$$\boldsymbol{\lambda}_X = \log\Big[\text{IFFT}\Big(\text{FFT}(\exp(\boldsymbol{\lambda}_{X_1} - \mu_{X_1})) \odot \text{FFT}(\exp(\boldsymbol{\lambda}_{X_2} - \mu_{X_2})))\Big)\Big] + \mu_{X_1} + \mu_{X_2}, \quad (12)$$

which expresses the log-probabilities $\boldsymbol{\lambda}_X$ in function of $\boldsymbol{\lambda}_{X_1}$ and $\boldsymbol{\lambda}_{X_2}$. It can still be computed in time $\mathcal{O}(M \log M)$ and avoids, at the same time, numerical stability issues by exponentiating $\boldsymbol{\lambda}_{X_i} - \mu_i$ instead of $\boldsymbol{\lambda}_{X_i}$ directly.

While using the fast log-conv-exp trick is necessary to scale computations in a numerically stable manner, describing operations on probability mass functions in the log-domain is rather cumbersome. Hence, we will, for the sake of clarity, describe PLIA$_t$ using probability vectors $\boldsymbol{\pi}$ (cf.. Sections 2.3, 3 and 4). We refer the reader to our implementation for the log-domain versions.

Another solution to the numerical instability of applying the FFT on probabilities was given in an application of the FFT to open-population [6] $N$-mixture models [27]. However, it has a major drawback when compared to the fast log-conv-exp trick: it relies on repeated applications of the traditional log-sum-exp trick within each of the $N \log N$ iterations of the FFT. This drawback prevents the use of optimised, off-the-shelf FFT algorithms and adds computational overhead. In contrast, we utilise the linearity of the FFT transform to provide an implementation-agnostic solution that works with tensorised representations.

## 2.3   Translational Invariance

In the previous sections we assumed that the first non-zero probability event of all probabilistic integers $X$ was the event $X = 0$, i.e. $L(X) = 0$. However, we can remove this assumption by characterizing a PMF $p_X$ not only by a vector of probabilities $\boldsymbol{\pi}_X$, but also by an integer $\beta_X = L(X)$ encoding the non-zero lower bound of the domain $\Omega(X)$. Indeed, we can write any PMF $p_X$ over an integer domain as the translation of a PMF $p_X^0$ whose first non-zero probability event is $X = 0$

$$p_X(x) = (\tau_{\beta_X} p_X^0)(x) = p_X^0(x + \beta_X). \quad (13)$$

Since the PMF $p_X^0$ can be represented by a probability vector $\boldsymbol{\pi}_X$ as in the previous sections, it follows that the PMF of any probabilistic integer can be characterised by such a vector and an integer $\beta_X$ that shifts the domain. The upper bound of the domain is not important for the characterisation of $p_X$ as it can be obtained via $U(X) = \beta_X + \dim(\boldsymbol{\pi}_X) - 1$, where $\dim(\boldsymbol{\pi}_X)$ is the dimension of the probability vector $\boldsymbol{\pi}_X$.

The inclusion of translations in our representation of PMFs is compatible with using convolutions to compute the PMF of the sum of two probabilistic integers because of the translational invariance of the convolution

$$\tau_k(f * g) = (\tau_k f * g) = (f * \tau_k g), \quad (14)$$

where $\tau_k$ denotes the translations by a scalar $k$. In general, $k$ can be real-valued, but for PLIA$_t$ we limit $k$ to integers. Using Equation 13, we can write the PMF of the sum of two probabilistic integers $X_1$ and $X_2$ with non-zero lower bounds $\beta_{X_1}$ and $\beta_{X_2}$ as

$$p_X = (p_{X_1} * p_{X_1}) = (\tau_{\beta_{X_1}} p_{X_1}^0 * \tau_{\beta_{X_2}} p_{X_1}^0) = (\tau_{\beta_{X_1}} \circ \tau_{\beta_{X_2}})(p_{X_1}^0 * p_{X_1}^0) \quad (15)$$

$$= \tau_{\beta_{X_1} + \beta_{X_2}}(p_{X_1}^0 * p_{X_1}^0). \quad (16)$$

This final equality shows that we can characterise the PMF $p_X$ for $X = X_1 + X_2$ by the following lower bound and probability vector

$$\beta_X = \beta_{X_1} + \beta_{X_2} \qquad \text{and} \qquad \boldsymbol{\pi}_X = F_M^{-1}\Big(F_M \boldsymbol{\pi}_{X_1} \odot F_M \boldsymbol{\pi}_{X_2}\Big). \quad (17)$$

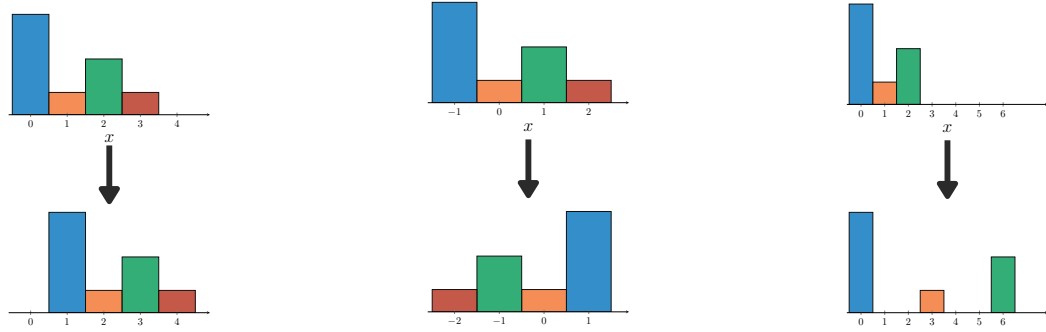

Figure 2: (Left) Adding a constant to a probabilistic integer simply means that we have to shift the corresponding histogram, shown here for $X' = X + 1$. (Middle) For the negation $X' = -X$, the bins of the histogram reverse their order and the negation of the upper bound becomes the new lower bound. (Right) For multiplication, here show the case $X' = 3X$ by inserting zero probability bins.

### 2.4 Formalising PLIA$_t$

PLIA$_t$ is concerned with computing the parametric form of the probability distribution of a linear integer arithmetic expression. It does so by representing random variables and linear combinations thereof as tensors whose entries are the log-probabilities of the individual events in the sample space of the random variable that is being represented. We define this formally as follows.

**Definition 2.1** (Probabilistic linear arithmetic expression). Let $\{X_1, \dots, X_N\}$ be a set of $N$ independent probabilistic integers with bounded domains. A probabilistic linear integer arithmetic expression $X$ is itself a bounded probabilistic integer of the form

$$X = \sum_{i=1}^{N} f_i(X_i), \tag{18}$$

where each $f_i$ denotes an operation performed on the specified random variables that can be either one of the operations specified in Section 3 as well as compositions thereof.

Note that operations within PLIA$_t$ are closed. That is, performing either of the operations delineated in Section 3 will again result in a bounded probabilistic integer representable as a tensor of log-probabilities and an off-set parameter indicating the value of the smallest possible event (cf. Section 2.3). In Section 4, PLIA$_t$ will also be provided with probabilistic inference primitives that allow it to compute certain expected values efficiently as well as to perform probabilistic branching.

Assuming all $f_i(X_i)$ are computable in polytime, we can also compute PLIA$_t$ expressions (Equation 18) in polytime in $N$. However, when computing PLIA$_t$ expressions recursively, the domain size of the random variables might grow super-polynomially – manifesting the #P-hard character of probabilistic inference.

## 3 Arithmetic on Integer-Valued Random Variables

The previous section introduced how PLIA$_t$ deals with the addition of two probabilistic integers. We discuss now five further operations: 1) addition of a constant, 2) negation, 3) multiplications by a constant, 4) integer division by a constant and 5) the modulo.

**Constant Addition.** The addition of a probabilistic integer $X$ and constant scalar integer $k$ forms a new probabilistic integer $X' = X + k$. Adding a scalar integer is equivalent to a translation of the distribution of $X$ (Figure 2, left). In other words, the lower bound and probability vector of $X'$ are given by

$$\beta_{X'} = \beta_X + k \qquad \text{and} \qquad \boldsymbol{\pi}_{X'} = \boldsymbol{\pi}_X. \tag{19}$$

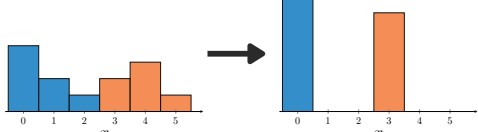 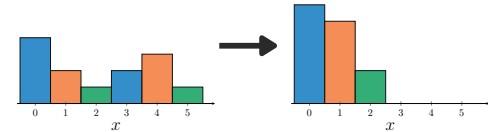

Figure 3: (Left) We show the histogram transformation for the integer division $X' = X/3$. The probability mass of three subsequent bins is accumulated in the bins for which $x \bmod 3 = 0$ and $x/3 \in \Omega(X)$. (Right) For the modulo $X' = X \bmod 3$, the only non-zero elements of $\Omega(X')$ are elements of the set $\{0, 1, 2\}$. The bins corresponding to these values then accumulate the probability masses of all other bins as indicated by the colors.

**Negation.** The negation $X' = -X$ of a probabilistic integer $X$ is equally straightforward to characterise. Taking a negation mirrors the probability distribution of $X$ around zero (Figure 2, middle). In terms of lower bound and probability vector, we get $\beta_{X'} = -(\beta_X + \dim(\boldsymbol{\pi}_X) - 1)$ and

$$\boldsymbol{\pi}_{X'}[x] = \begin{cases} \boldsymbol{\pi}_{X'}[x] = \boldsymbol{\pi}_X[\dim(\boldsymbol{\pi}_X) - x - 1], & \text{if } 0 \le x < \dim(\boldsymbol{\pi}_X), \\ 0, & \text{otherwise.} \end{cases} \tag{20}$$

respectively. That is, the lower bound of $X'$ is equal to the negated upper bound of $X$ while the probability vector is flipped, taking into account that probability vectors have to start at 0.

**Constant Multiplication.** For the multiplication $X' = X \cdot k$ of a probabilistic integer $X$ with a scalar integer $k$, we assume, without loss of generality, that $k \ge 0$. Multiplication by a scalar is then characterised as

$$\beta_{X'} = \beta_X \cdot k \quad \text{and} \quad \boldsymbol{\pi}_{X'}[x] = \begin{cases} \boldsymbol{\pi}_X[\frac{x}{k}], & \text{if } x \bmod k = 0 \text{ and } 0 \le \frac{x}{k} < \dim(\boldsymbol{\pi}_X), \\ 0, & \text{otherwise.} \end{cases} \tag{21}$$

Intuitively, only multiples of $k$ get a non-zero probability equal to the probability of that multiple in $\boldsymbol{\pi}_X$. The lower bound of $X'$ is also immediately given by multiplying the lower bound of $X$ by $k$. In other words, we obtain $\boldsymbol{\pi}_{X'}$ by inserting $k - 1$ zeros between every two subsequent entries of $\boldsymbol{\pi}_X$ (Figure 2, right). The case $k < 0$ is obtained by first negating $X$.

**Integer Division and Modulo.** For the case of integer division $X' = X/k$ and the modulo operation $X' = X \bmod k$, the probability distribution of $X'$ can be obtained by adequately accumulating probability mass from events in $\Omega(X)$. We demonstrate these operations by example in Figure 3 and refer the reader to Appendix A for the formal description.

## 4 Probabilistic Inference Primitives

### 4.1 Computing Expected Values

$\text{PLIA}_t$ supports the exact computation of two different forms of expected values. The first is a straightforward expectation of a probabilistic integer $X$, given by weighing each element of $\Omega(X)$ with its probability

$$\mathbb{E}[X] = \sum_{x \in \Omega(X)} x \cdot \boldsymbol{\pi}_X[x - \beta_X]. \tag{22}$$

The second is computing the expectation of a linear comparative expression of probabilistic integers. Such a comparison can be an equality, inequality or negated equality. We only consider the equality and strictly larger inequality as the other cases follow from them. The strict inequality can itself always be reduced to an inequality with respect to zero and hence comprises a sum over all domain elements below zero

$$\mathbb{E}[\mathbb{1}_{X<0}] = \sum_{x \in \Omega(X): x < 0} \boldsymbol{\pi}_X[x - \beta_X]. \tag{23}$$

Similarly, the computation of the expected value of an equality comparison can always be reduced to a comparison to zero. Hence, the expected value is computable by simple indexing

$$\mathbb{E}[\mathbb{1}_{X=0}] = \boldsymbol{\pi}_X[-\beta_X]. \tag{24}$$

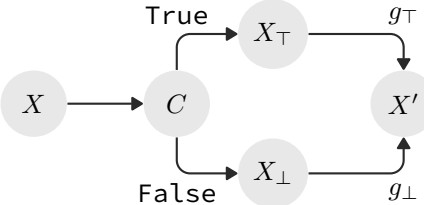

Figure 4: Control flow diagram for probabilistic branching. The branching condition is probabilistically true and induces a binary random variable $C$. In each of the two branches we then have two conditionally independent random variables $X_\top$ and $X_\bot$ to which the functions $g_\top$ and $g_\bot$ are applied in their respective branches. The probabilities of $X'$ are then given by the weighted sums of the probabilities of $g_\top(X_\top)$ and $g_\bot(X_\bot)$ (Equation 29).

As computing expected values is no harder than computing the sum of the elements of a vector, we can conclude that we can compute these expected values in $\mathcal{O}(\dim(\boldsymbol{\pi}_X))$. By using prefix sums [18] and harnessing parallel compute on GPUs, the complexity can further be reduced to $\mathcal{O}(\log \dim(\boldsymbol{\pi}_X))$

### 4.2 Probabilistic Branching

Consider an if-then-else statement with condition $c(x) = (f(x) \bowtie 0)$, where $f$ is a composition of the functions introduced in Section 3 and $\bowtie \in \{<, \leq, =, >, \geq, \neq\}$. Furthermore, $x$ belongs to the domain $\Omega(X)$ of a probabilistic integer $X$. In the case of $c(x)$ being true, a function $g_\top$ is executed. If $c(x)$ is false, another function $g_\bot$ is executed instead. We assume that both $g_\top$ and $g_\bot$ are again linear arithmetic functions expressible in PLIA$_t$ (Section 2.4).

The if-then-else statement defines a new probabilistic integer $X'$ by combining both of its branches (Figure 4). These branches depend on $X$ which itself influences a binary random variable $C$ that represents the probabilistic condition of the if-then-else statement. To be precise, the PMF $p_{X'}$ is given by the decomposition

$$p_{X'}(x') = p_{X'|C}(x' \mid \top) \cdot p_C(\top) + p_{X'|C}(x' \mid \bot) \cdot p_C(\bot), \tag{25}$$

where $p_{X'|C}$ is the conditional PMF of $X'$ given $C$. The true branch gives rise to a probabilistic integer $X_\top$ with probability distribution

$$p_{X_\top}(x) = p_{X|C}(x \mid \top) = \frac{p_{C|X}(\top \mid x) p_X(x)}{p_C(\top)} = \frac{\mathbb{1}_{c(x)} \boldsymbol{\pi}_X[x - \beta_X]}{p_C(\top)}. \tag{26}$$

If $C = \top$, then $X'$ is given by an application of $g_\top$ on the instances $x \in \Omega(X)$ that satisfy $c(x)$. Consequently, by applying $g_\top$, we find that

$$p_{X'|C}(x' \mid \top) = p_{g_\top(X_\top)}(x'). \tag{27}$$

With the right-hand side of Equation 26, we now know how to obtain the probability vector for $X_\top$. Using Equation 27 and the operations from Section 3, we can then compute the probability vector $\boldsymbol{\pi}_{g_\top(X_\top)}$, as well as the lower bound $\beta_{g_\top(X_\top)}$. Also note that $p_C(\top)$ is nothing but an expected value as described by Equation 23 or Equation 24. Analogously, we obtain for the false branch that

$$p_{X_\bot}(x) = \frac{(1 - \mathbb{1}_{c(x)}) \boldsymbol{\pi}_X[x - \beta_X]}{p_C(\bot)} \quad \text{and} \quad p_{X'|C}(x' \mid \bot) = p_{g_\bot(X_\bot)}(x'). \tag{28}$$

By plugging the expressions for $p_{X'|C}(x' \mid \top)$ and $p_{X'|C}(x' \mid \bot)$ into Equation 25 we find that

$$p_{X'}(x') = p_{g_\top(X_\top)}(x') \cdot p_C(\top) + p_{g_\bot(X_\bot)}(x') \cdot p_C(\bot), \tag{29}$$

which are all quantities computable using either probabilistic linear arithmetic operations or expected values thereof.

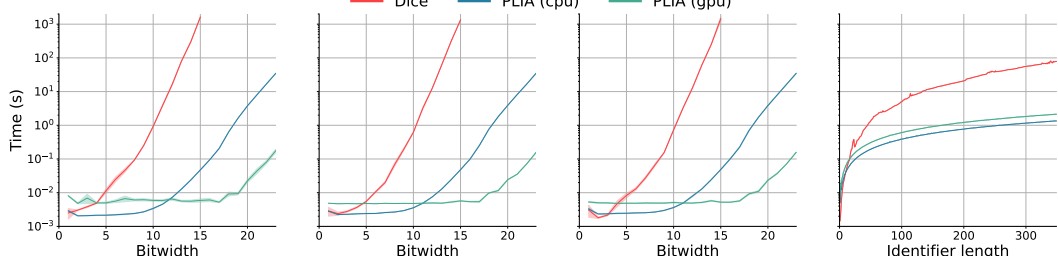

Figure 5: We plot the runtime of Dice [5] and $\text{PLIA}_t$ against the domain size of the problems. From left to right, we have $\mathbb{E}[X_1 + X_2]$, $\mathbb{E}[\mathbb{1}_{X_1+X_2<0}]$, $\mathbb{E}[\mathbb{1}_{X_1+X_2=0}]$ and probabilistic Luhn. All four plots share the same y-axis on the very left, which is in log-scale. Following the experimental protocol of Cao et al. [5], we report average runtimes for every integer on the x-axis, both bitwidths and identifier lengths. No significant deviations from the mean were found.

## 5 Experiments

We first compare $\text{PLIA}_t$ to the state of the art in probabilistic integer arithmetic [5] in terms of inference speed (Section 5.1). These experiments were performed using an Intel Xeon Gold 6230R CPU @ 2.10GHz, 256GB RAM for CPU experiments and an Nvidia TITAN RTX (24GB) for GPU experiments. In Section 5.2, we then illustrate how $\text{PLIA}_t$ fares against the state of the art in neurosymbolic AI [14, 33]. These experiments were performed using an Nvidia RTX 3080 Ti (12GB). We implemented $\text{PLIA}_t$ in TensorFlow [1] using the Einops library [30]. This implementation is open-source and available at `https://github.com/ML-KULeuven/probabilistic-arithmetic`

### 5.1 Exact Inference with Probabilistic Integers

The work of Cao et al. [5] exploits the structural properties and symmetries of integer arithmetic by proposing general encoding strategies for an arithmetic expression of probabilistic integers as logical circuits. That is, binary decision diagrams [4] obtained via knowledge compilation [7]. This strategy allows them to avoid redundant calculations and repetition, leading to improved scalability over more naive encodings [8, 13].

We compare $\text{PLIA}_t$ and Cao et al.'s inference algorithm on four of their benchmark problems. In the first three benchmarks, expected values of the sum of two random variables need to be computed. Concretely, the expectations $\mathbb{E}[X_1 + X_2]$ (cf. Equation 22), $\mathbb{E}[\mathbb{1}_{X_1+X_2<0}]$ (cf. Equation 23) and $\mathbb{E}[\mathbb{1}_{X_1+X_2=0}]$ (cf. Equation 24).

As a fourth benchmark we use a probabilistic version of the Luhn checksum algorithm [20], which necessitates summation of two probabilistic integers, negation, addition of a constant, multiplication by a constant, the modulo operations, as well as probabilistic branching. We provide further details on the Luhn algorithm in general and the encoding of its probabilistic variant in $\text{PLIA}_t$ in Appendix C.

As the probabilistic Luhn algorithm takes as input an identifier consisting of a sequence of probabilistic integers with domain $\{0, \ldots, 9\}$, we can increase the problem size by increasing the length of this sequence. For the other three benchmarks we vary the problem size by varying the domain size of the probabilistic integers in terms of their *bitwidth*. That is, a bitwidth of $i \in \mathbb{N}$ indicates that we consider probabilistic integers ranging from 0 up until $2^i - 1$, increasing the problem size exponentially in terms of $i$. In our experimental evaluation (Figure 5), we measure the time it took for $\text{PLIA}_t$ and Cao et al.'s method to terminate for varying problem sizes. The measured time includes all computational overhead inherent to each method, such as the construction of computational graphs and compilation time. Each method is also profiled in terms of memory, which we discuss further in Appendix B. Note that Cao et al.'s method is denoted by "Dice", the probabilistic programming language in which it was implemented.

For the first three benchmarks, we observe that $\text{PLIA}_t$ easily scales to probabilistic integers with a domain size of $2^{24}$ on both the CPU and GPU as the highest runtime reached is less than 100 seconds on the CPU and approximately 1 second on the GPU. The similarity of the curves is due to the fact

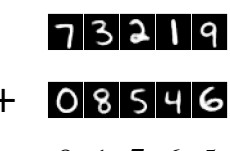

$$+ \quad \boxed{0}\boxed{8}\boxed{5}\boxed{4}\boxed{6}$$

$$= \quad 8 \ 1 \ 7 \ 6 \ 5$$

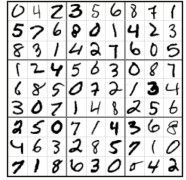

Figure 6: (Left) Example of an MNIST addition data point, consisting of two numbers given as a series of MNIST digits and an integer. The integer is the sum of the two numbers and constitutes the label of the data point. (Right) Data point from the visual sudoku data set, consisting of a $9 \times 9$ grid filled with MNIST digits. Data points are labeled with a Boolean value indicating whether the integers underlying the MNIST digits satisfy the constraints of sudoku.

that the run time is dominated by computing the probability vector $\boldsymbol{\pi}_{X_1+X_2}$ and not so much by computing the actual expected value.

In contrast, Dice, which only runs on CPU, already reaches a runtime of approximately 1000 seconds for integers with domain size $2^{15}$, where PLIA$_t$ only takes around $10^{-1}$ and $10^{-2}$ seconds on the CPU and GPU, respectively. This is a rather considerable improvement in the order of $\mathbf{10^5}$. Dice can outperform PLIA$_t$ on the GPU (Figure 5, bitwidth smaller strictly below 5) due to the computational overhead of running on the GPU. However, much of this overhead can be avoided by running PLIA$_t$ on the CPU for smaller domain sizes, where it performs on par or better than Dice (Figure 5, bitwidth smaller strictly below 5).

On the probabilistic Luhn benchmark (Figure 5, extreme right) we observe that both methods exhibit similar linear scaling behaviors. However, the use of tensors as representations instead of logical circuits does result in a significant improvement in terms of run time in the order of $10^2$ for the longest sequences of length 350.

## 5.2 Neurosymbolic Learning

For the comparison of PLIA$_t$ to neurosymbolic systems, we use two standard benchmarks from the literature: MNIST addition [21] and visual sudoku [2]. The common idea for both is to train neural networks to classify MNIST digits while only having access to distant supervision (Figure 6).

As an MNIST classifier outputs a distribution over the integers $\{0, \ldots, 9\}$, we can readily encode these predictions as probabilistic integers and enforce the constraints given by the two problems using the arithmetic operations developed in Section 2 and Section 3. We refer the reader to Appendix D (MNIST addition) and Appendix E (visual sudoku) for details on the encodings.

In the experimental evaluation we compare PLIA$_t$, which uses exact probabilistic inference, to one other exact method, DeepProbLog (DPL) [21, 22], and two approximate methods, Scallop [14] and A-NeSI [33]. Similar to Dice (Section 5.1), DPL also relies on expensive knowledge compilation in order to obtain the distribution of the sum of two probabilistic integers. Essentially, it performs explicit enumeration as illustrated in Figure 1. Scallop approximates this explicit enumeration by only considering the top-$k$ most likely solutions of a problem. The approximation of A-NeSI is based on optimising a neural surrogate model for the combinatorial problem. While this model sidesteps the computational complexity encountered by DPL and Scallop to a certain degree, training the surrogate model becomes prohibitively expensive for larger problems.

We compare the different methods along two dimensions, being prediction accuracy and training time. Specific details on training, e.g. neural architectures and hyperparameters, can be found in Appendix F. We report the statistics in Table 1, where we use numbers consisting of $N \in \{2, 4, 15, 50\}$ digits for the MNIST addition benchmark and grid size $G \in \{4, 9\}$ for the visual sudoku benchmark. We see that PLIA$_t$ significantly outperforms the other methods, both exact as well as approximate, in terms of training times. The difference is particularly apparent on the MNIST addition benchmark, where no other method was able to scale up to $N = 50$ without timing out. The reported accuracies also show that this advantage of training time for PLIA$_t$ with respect to the other methods does not come at the cost of predictive performance of the learned neural networks.

Table 1: In the upper part part we report median test accuracies over 10 runs for the MNIST addition and the visual sudoku benchmarks for varying problem sizes and different neurosymbolic frameworks. Sub- and superscript indicate the 25 and 75 percent quantiles, respectively. In the lower part we report the training times in minutes, again using medians with 25 and 75 percent quantiles. We set the time-out to 24 hours (1440 minutes).

| Method | MNIST Addition | | | | Visual Sudoku | |
|---|---|---|---|---|---|---|
| | $N = 2$ | $N = 4$ | $N = 15$ | $N = 50$ | $G = 4$ | $G = 9$ |
| | **Accuracy** | | | | **Accuracy** | |
| DPL | $94.20^{+2.04}_{-0.16}$ | T/O | T/O | T/O | T/O | T/O |
| Scallop | $95.40^{+0.40}_{-0.08}$ | $90.88^{+0.48}_{-0.48}$ | T/O | T/O | $74.50^{+2.00}_{-0.00}$ | T/O |
| A-NeSI | $95.40^{+0.60}_{-0.24}$ | $92.40^{+0.64}_{-0.56}$ | $74.71^{+2.94}_{-1.27}$ | T/O | $85.50^{+2.00}_{-0.00}$ | $61.50^{+5.00}_{-2.50}$ |
| $PLIA_t$ | $95.88^{+0.28}_{-0.16}$ | $91.60^{+0.72}_{-0.16}$ | $79.09^{+1.21}_{-2.12}$ | $43.00^{+5.00}_{-3.00}$ | $85.00^{+3.00}_{-4.50}$ | $63.00^{+1.00}_{-1.00}$ |
| | **Timings (minutes)** | | | | **Timings (minutes)** | |
| DPL | $88.21^{+15.58}_{-1.89}$ | T/O | T/O | T/O | T/O | T/O |
| Scallop | $22.62^{+0.64}_{-0.06}$ | $50.41^{+6.46}_{-0.17}$ | T/O | T/O | $3.90^{+0.01}_{-0.02}$ | T/O |
| A-NeSI | $41.02^{+8.54}_{-2.79}$ | $53.62^{+6.40}_{-1.76}$ | $714.55^{+27.66}_{-8.17}$ | T/O | $30.67^{+0.98}_{-0.17}$ | $134.86^{+31.01}_{-8.13}$ |
| $PLIA_t$ | $1.97^{+0.02}_{-0.01}$ | $2.44^{+0.04}_{-0.04}$ | $7.85^{+0.64}_{-0.19}$ | $11.98^{+0.68}_{-0.04}$ | $0.58^{+0.03}_{-0.02}$ | $6.39^{+0.01}_{-0.01}$ |

## 6 Conclusion and Future Work

We introduced $PLIA_t$, an efficient, differentiable and hyperparameter-free framework for linear arithmetic over integer-valued random variables. The efficiency of $PLIA_t$ is due to representing probabilistic integers as tensors and exploiting the FFT for computing the sum of two probabilistic integers. Compared to state-of-the-art methods for inference [5] and learning [33] the concepts underlying $PLIA_t$ are surprisingly simply: a tensorised calculus and the fast Fourier transform. This simple yet elegant approach has led to improvements in inference and learning times in the order of $10^5$ and $10^2$, respectively. We attest this advantage to $PLIA_t$'s formulation in terms of fundamental concepts shared across modern machine learning. As such, any algorithmic or hardware improvements will immediately improve $PLIA_t$'s performance. For instance, incorporating recent advancements for computing FFTs on GPUs [11] would directly benefit $PLIA_t$'s efficiency.

In future work we envisage to extend the probabilistic inference primitives of $PLIA_t$ to a full-fledged neuroprobabilistic programming language, resulting in a user-friendly interface to an efficient neurosymbolic reasoning engine. In this regard, we deem ideas from the satisfiability modulo theory literature [3] as important. Specifically, probabilistic [17, 25] and neural [9] extensions thereof, as well as novel formula representations [10, 24].

## Acknowledgements

This work was supported by the KU Leuven Research Fund (C14/18/062 and iBOF/21/075) and the Flemish Government under the "Onderzoeksprogramma Artificiële Intelligentie (AI) Vlaanderen" programme. It also received funding from the Research foundation - Flanders project "Neural Probabilistic Logic Programming" (G097720N). PZD is supported by the Wallenberg AI, Autonomous Systems and Software Program (WASP) funded by the Knut and Alice Wallenberg Foundation.

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

# Appendix

## A   Formal Description of Integer Division and Modulo

**Integer Division.** Performing a scalar integer division of the random variable $X$ by $k$ yields the random variable $X' = \frac{X}{k}$ with lower bound $\beta_{X'} = \frac{\beta_X}{k}$ and probability vector

$$\boldsymbol{\pi}_{X'}[x] = \begin{cases} \sum_{i=k\cdot x}^{k\cdot x+k-1} \boldsymbol{\pi}_X[i], & \text{if } 0 \le x < \frac{\dim(\boldsymbol{\pi}_X)}{k}, \\ 0 & \text{otherwise,} \end{cases} \tag{30}$$

where we assume that $L(X) \bmod k = 0$ and $U(X) + 1 \bmod k = \dim(\boldsymbol{\pi}_X) \bmod k = 0$. This is a weak assumption as it can easily be obtained via zero-padding. The intuition is then that the probability vector $\boldsymbol{\pi}_{X'}$ of an integer division of $X$ by $k$ accumulates probability mass of $k$ sequential entries of $\boldsymbol{\pi}_X$ (Figure 3, left).

**Modulo Operation.** For the modulo of a random variable by a constant, i.e. $X' = X \bmod k$, we have the lower bound $\beta_{X'} = 0$ and the probability vector is

$$\boldsymbol{\pi}_{X'}[x] = \begin{cases} \sum_{i=0}^{\frac{\dim(\boldsymbol{\pi}_X)}{k}} \boldsymbol{\pi}_X[x + k \cdot i], & \text{if } 0 \le x < k, \\ 0, & \text{otherwise,} \end{cases} \tag{31}$$

where we again assumed that $L(X) \bmod k = 0$ and $U(X) + 1 \bmod k = \dim(\boldsymbol{\pi}_X) \bmod k = 0$. In other words, the probability vector $\boldsymbol{\pi}_{X'}$ of the modulo of a random variable $X$ accumulates the probability mass of $\boldsymbol{\pi}_X$ by multiples of $k$ (Figure 3, right).

## B   Studying the Memory Consumption of PLIA$_t$

It is well-known that the FFT has, apart from the $N \log N$ runtime complexity, also a $N \log N$ theoretical space complexity, again in contrast to a $N^2$ space complexity of the naive approach. Moreover, we performed an additional empirical comparison between Dice [5] and PLIA$_t$ with respect to memory allocation (Figure 7). We found that Dice was only more efficient for smaller problem sizes due to the memory overhead of TensorFlow on the GPU. However, from problem sizes of bitwidth higher than 13, Dice overtook PLIA$_t$ in memory usage and we confirmed the better scaling behaviour of PLIA$_t$. Hence, PLIA$_t$ not only outshines the current state of the art in terms of runtime but also in terms of memory usage. Memory usage of PLIA$_t$ on the CPU could not be properly profiled due to Python's limited functionality, but there is no reason to suspect it would be significanlty different from the GPU memory usage.

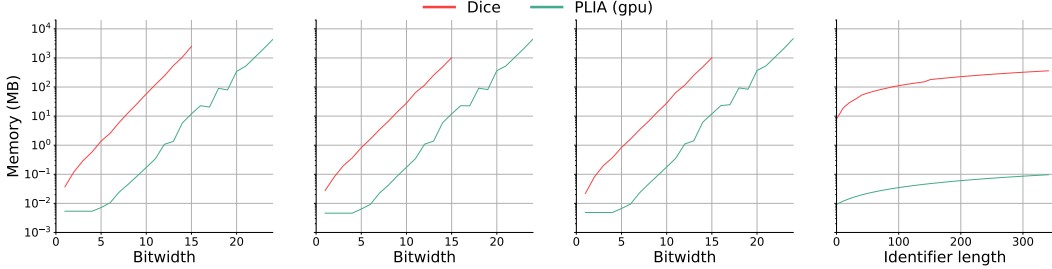

Figure 7:   The memory usage in MB plotted against the domain size of each of the problems discussed in Section 5.1. From left to right, we have $\mathbb{E}\left[X_1 + X_2\right]$, $\mathbb{E}\left[\mathbb{1}_{X_1+X_2<0}\right]$, $\mathbb{E}\left[\mathbb{1}_{X_1+X_2=0}\right]$ and probabilistic Luhn. All four plots share the same y-axis on the very left, which is in log-scale. Memory measurements were consistently identical across multiple different runs.

```python
1  def luhn_checksum(identifier):
2      check = 0
3      for i, digit in enumerate(identifier):
4          if i % 2 == len(identifier) % 2:
5              ite_digit = ifthenelse(
6                  digit,
7                  lt=5,
8                  tbranch=lambda x: 2 * x,
9                  fbranch=lambda x: 2 * x - 9,
10             )
11             check = check + ite_digit
12         else:
13             check = check + digit
14         check = check % 10
15     return check
```

Algorithm 1: Probabilistic Luhn algorithm written in Python using our `plia` library.

## C  Luhn Program

The traditional Luhn algorithm [20] computes a check digit from a given sequence of integers. If the check digit is correct, then it means that there are no single-entry mistakes in the sequence. Such errors often arise, for example, when transcribing credit card information. Hence, Luhn is often used in this context to quickly identify mistyped numbers.

We provide an implementation of the probabilistic Luhn algorithm using PLIA$_t$ in Algorithm 1. During its execution, the algorithm sequentially constructs the check digit by iterating over the given sequence. At the start of every step, a deterministic if-then-else statement checks whether the current sequence has the same binary polarity as the length of the total sequence to be checked (Line 4). The first argument of the `ifthenelse` function is the probabilistic integer on which to perform the test. The second argument gives the test. For Luhn we have `digit<5`. The two lambda functions then tell us which function to perform on the random variable in each of the two branches. The function outputs a probabilistic integer which we add to the `check` probabilistic integer.

## D  MNIST Encoding

Let $n_1$ and $n_2$ be two probabilistic integers. The MNIST addition problem is then usually encoded in one of two different ways. The first approach simply directly constructs the joint distribution over all possible integers. The second approach avoids this explicit constructions by computing conditional distributions for intermediate computations. In other words, the first approach constructs a lookup table for the sum of two integers while the second encoding gives us a probabilistic version of the sum of two integers via a carry. Naturally, the second approach is able to scale to larger numbers and we also use it in our experiment for all methods. We give the implementation for both encodings in Algorithmns 2 and Algorithm 3.

## E  Visual Sudoku Probabilistic Binary Encoding

For the problem of solving a visual sudoku [2], the input is a $G \times G$ sudoku puzzle filled with handwritten digits (Figure 6, right). The task itself is to predict whether such a given sudoku puzzle is valid, meaning the values of the digits satisfy the various sudoku constraints. For $G = 4$, the constraints to satisfy are that every row and every column must have different entries. In case $G = 9$, all nine inner $3 \times 3$ grids of the puzzle must also have different entries. Again, because the digits are handwritten, a neural network has to classify the digits into probabilistic integers from $0$ to $9$.

```
1  def sum_numbers(
2      probs_1,
3      probs_2,
4      digits_per_number
5  ):
6      number1 = 0
7      number2 = 0
8      for i in range(digits_per_number):
9          number1 += PInt(probs_1[i], 0) * 10 ** i
10         number2 += PInt(probs_2[i], 0) * 10 ** i
11     result = number1 + number2
12     return result
```

Algorithm 2: Computing the sum of two numbers by explicitly constructing the joint distributions of the resulting digit. PInt is the `plia` primitive to construct probabilistic integers.

```
1  def sum_numbers(
2      probs_1, probs_2,
3      digits_per_number
4  ):
5      carry = 0
6      result = []
7      for i in range(digits_per_number):
8          sum_digit = PInt(probs_1[i], 0)
9                    + PInt(probs_2[i], 0)
10                   + carry
11         result.append(sum_digit % 10)
12         carry = sum_digit // 10
13     result.append(carry)
14     return result
```

Algorithm 3: Computing the sum of two probabilistic digits by computing conditional probabilities of the next digit in the resulting sum given the current digits by means of a carry.

Consequently, this task requires both solving the now probabilistic combinatorial problem together with training the neural classifiers. Note that this problem is far from trivial, as there are $10^{G \cdot G}$ possible ways to fill in a $G \times G$ sudoku puzzle.

Checking whether a sudoku puzzle is correct requires checking if the sudoku constraints are satisfied. While there are many encodings of these constraints, $\text{PLIA}_t$ needs an encoding in terms of linear arithmetic operations. The encoding we used is as follows. Every grid cell is associated with 10 binary probabilistic integers, leading to $G \cdot G \cdot 10$ such integers for the whole puzzle. Intuitively, the probability of the $k^{\text{th}}$ binary integer of a grid cell being equal to 1 should be interpreted as the probability that the grid cell contains the number $k$. We represent these binary integers as a 3-tensor $B_{ijk}$, where $i$ indexes a row of the grid, $j$ indexes a column and $k$ is the $k^{\text{th}}$ binary digit of grid cell $(i, j)$. With this representation, expressing that a row $i$ should only contain different digits is equivalent to obtaining a value of 1 after summing out the index $j$ from $B_{ijk}$. Concretely, that is

$$\sum_{j=1}^{G} B_{ijk} = 1, \qquad \forall i \in \{1, \ldots, G\} \text{ and } \forall k \in \{1, \ldots, 10\}. \tag{32}$$

Indeed, for a fixed value of $i$, i.e. for a single row $i$, there are 10 constraints to satisfy, where each one excludes double presences of one of the 10 possible digits. The constraint that each column only

contains different digits is completely analogous to Equation 32, but instead summing over the row index $i$.

In case $G = 9$, there are the additional constraints that the inner 9 blocks of the puzzle also have no repeating entries. The intuition is similar to the other two constraints; we sum over all grid cells of a block for each of the binary integers and should obtain 1. If we index the 9 inner blocks as a $3 \times 3$ grid with coordinates $(l, m)$, then the equations

$$\sum_{i=3 \cdot l}^{3 \cdot (l+1)} \sum_{j=3 \cdot m}^{3 \cdot (m+1)} B_{ijk} = 1, \qquad \forall l, m \in \{0, 1, 2\} \text{ and } \forall k \in \{1, \ldots, 10\}, \tag{33}$$

is true if and only if each of the inner blocks has different digits in all of its grid cells.

Bringing everything together, $\text{PLIA}_t$ approximates the probability of a sudoku puzzle being correct by computing the product of the probabilities of all the above linear constraints. Note that this approach computes each of the constraints as independent of one another, allowing for the probability of each constraint to be computed in parallel. Because $\text{PLIA}_t$ is based on tensor manipulations, our implementation can and does exploit this parallelism. Additionally, note that $\text{PLIA}_t$ works with log-probabilities in practice, which avoids the problem of numerical underflow often arising when taking long products of probabilities.

**Choice of Encoding for Comparisons.** The other methods we compare to do not necessarily need to use the same binary linear arithmetic encoding as they are not restricted to linear arithmetic. Instead, their provided implementations directly encode every grid cell as a categorical random variable with 10 outcomes, one for every possible digit. Expressing that a row, column or block of grid cells only contains different digits is then done by comparing all pairs of categorical variables in the row, column or block. Each of these pairs then has to be different for the constraint to hold. To again provide a fair comparison, we tested both encodings for Scallop and A-NeSI and took the best performing one. In both cases, the categorical encoding gave better runtime results.

## F   Details on Training the Neurosymbolic Methods

Since both the MNIST and visual sudoku task involve the classification of MNIST images, we use the same traditional LeNet [19] in both cases. Next we discuss the hyperparameters involved in training.

The optimal parameters for $\text{PLIA}_t$ and Scallop were found by performing a grid search on a held-out validation set. The main hyperparameters important for all methods used for neurosymbolic learning are the learning rate and the number of epochs to train. For our grid search, we considered values $\{0.001, 0.005, 0.0001\}$ for the learning rate and $\{5, 10, 15, 20, 50, 100, 200\}$ for the number of epochs. Scallop and $\text{PLIA}_t$ both consistently performed best with a learning rate of $0.001$ in all experiments. Both Scallop and $\text{PLIA}_t$ use Adam [16] as optimiser. The number of epochs varies per experiment. On the MNIST task, $\text{PLIA}_t$ needed 10, 15, 100 and 200 epochs for $N = 2$, $N = 4$, $N = 15$ and $N = 50$, respectively. For Scallop, $N = 2$ and $N = 4$ ran optimally when training for 5 and 10 epochs, respectively.

For Scallop, with its top-$k$-based approximate inference, it is additionally important to choose a good value for $k$. If $k$ is too small, then the approximation quality is poor. If $k$ is too large, then it becomes prohibitively expensive to compute the approximation. Inspired by the implementation of Scallop, the possible values for $k$ were $\{1, 3, 5, 6, 7\}$. For the MNIST addition task, in all cases that did not time out, $k = 3$ was optimal as it gave state-of-the-art performance and the quickest runtimes. The visual sudoku problem benefitted from a higher value of $k$, having an optimal value of $k = 6$. No value of $k$ allowed Scallop to run on a $9 \times 9$ sudoku.

In stark contrast to $\text{PLIA}_t$ A-NeSI has a multitude of hyperparameters to be set. We used the set of parameters provided in the original paper [33] and its implementation that were already used there to perform the experiments on MNIST addition and visual sudoku.

