# OpenReview forum: "A Fast Convoluted Story: Scaling Probabilistic Inference for Integer Arithmetics"
_NeurIPS.cc/2024/Conference — NeurIPS 2024 poster_

### Official Review · Reviewer_DnxY · 2024-07-12

**Soundness:** 3
**Presentation:** 4
**Contribution:** 2
**Rating:** 6
**Confidence:** 2

**Summary:**

The authors study the problem of probabilistic inference's scalability in the integer arithmetic setting. Their method is based on the representation of integer-valued random variables probability mass functions (PMFs) as vectors and the observation that PMFs of different operations applied on said random variables can be expressed as convolutions of the PMFs vectors by leveraging Fast Fourier Transforms.

**Strengths:**

The method and the mathematical background underlying the proposed approach for probabilistic inference over integer-valued random variables are well-detailed.
The Fast Log-Conv-Exp trick and related computational aspects are properly discussed.
The PLIA framework and its implementation seem fairly simple, allowing the authors to scale probabilistic inference over larger integer problems and neurosymbolic learning surpassing previous state-of-the-art approaches.

**Weaknesses:**

- The experimental part is a bit limited and could be extended to show more clearly potential and limitations of the method

**Questions:**

- More test to understand the limits of the approach in terms of scalability can be useful. How far can you push the size of MNIST addition by allowing, e.g. 1 hour computational time?

**Limitations:**

-

---

> ### Author Rebuttal · Authors · 2024-08-06
>
> We thank the reviewer for their supportive review. Please allow us to address your experimental concern below.
>
> Our experimental evaluation tackled two of the most prominent neurosymbolic benchmarks, where PLIA$_t$ outperformed both exact and approximate state-of-the-art methods for neurosymbolic learning. It demonstrates that the field of neurosymbolic AI is in need of more challenging benchmarks, as we have shown that it is now possible to solve both MNIST addition and visual sudoku quickly and accurately. The goal of our paper is to scale probabilistic inference on linear integer arithmetic, which we illustrate by outperforming the state-of-the-art in inference (exact method) by 100 000x and the state of the art in learning (approximate method) by 100x in terms of run time. Hence, we fully agree it is high time for more challenging benchmarks, yet proposing new benchmarks is not the aim of this paper.
>
> Further scaling the MNIST addition task to more digits is limited by the size of the dataset, as there are only 60 000 images available for training and 10 000 for testing. Each of these images can only be used once in the construction of the various sums, meaning increasing the number of digits leads to an overall reduction of available sums for training and testing. For example, our case of $N = 50$ requires 100 images to construct a single sum data sample. Consequently, the test dataset only contains 100 samples, which was the lowest number of samples we were still comfortable with to compute test statistics over. We did run learning experiments with $N = 100$ and even $N = 1000$, but we do not deem their test statistics faithful enough to be reported.

---

> > ### Comment · Reviewer_DnxY · 2024-08-09
> >
> > Thanks for your comments

---

### Official Review · Reviewer_GGzh · 2024-07-12

**Soundness:** 3
**Presentation:** 2
**Contribution:** 3
**Rating:** 5
**Confidence:** 4

**Summary:**

The paper presents a framework, PLIA_t, to solve the generally intractable problem of probabilistic inference using the fast Fourier transform (FFT) for integer-valued random variables. The paper shows how to use the log-sum-exp trick to solve the numerical stability issue in the FFT setting and defines arithmetic operations on integer-valued random variables. It also defines how to compute expected values and how to handle probabilistic branching. Experiments show that the gains can be realised in an implementation for the given examples.

**Strengths:**

- The paper tackles a central problem, probabilistic inference, from a new perspective to the best of my knowledge.
- The results are promising.

**Weaknesses:**

- Even though the paper plans to introduce PLIA_t as a new framework, the framework itself remains rather opaque regarding its inputs, outputs, representations, algorithms, or whatever they consider part of PLIA_t. It almost reads as if a section is missing between Section 4 and 5. As such, I am also not so sure about the reproducibility of the experiments. (I have formulated a corresponding question. This is one of the main reasons for my overall score that I am willing to revise given a convincing rebuttal.)

**Questions:**

- Can you please characterise / define PLIA_t formally as a framework in the rebuttal?

**Limitations:**

Yes.

---

> ### Author Rebuttal · Authors · 2024-08-06
>
> We thank the reviewer for their time and effort, and for their suggestion of formalizing PLIA$_t$. For the camera-ready version we propose to add the following subsection at the end of the Section 2:
>
> ### 2.4. Formalizing PLIA$_t$
>
> **Definition** *(probabilistic linear arithmetic expression)
> Let  $\{X_1, …X_N\}$ be a set of $N$ independent integer-valued random variables with bounded domains. A probabilistic linear integer arithmetic expression X is a bounded integer-valued random variable of the form
> $$
> X = \sum_{i=1}^{N} f_i (X_i).
> $$
> In the equation above the functions $f_i$ denote operations performed on the specified random variables that can be either one of the operations specified in Section 3 as well as compositions thereof.*
>
> PLIA$_t$ is concerned with computing the parametric form of the probability distribution of a linear integer arithmetic expression. It does so by representing random variables and linear combinations thereof (cf. Definition of probabilistic linear arithmetic expressions) as tensors whose entries are the log-probabilites of the individual events in the sample space of a the random variable that is being represented. Note that operations within PLIA$_t$ are closed. That is, performing either of the operations delineated in Section 3 will again result in a bounded probabilistic integer representable as a tensor of log-probabilities and an off-set parameter indicating the value of the smallest possible event (cf. Section 2.3). In Section 4 we also equip PLIA$_t$ with probabilistic inference primitives that allow it to compute certain expected values efficiently as well as to perform probabilistic branching.
>
>
> <END OF SECTION 2.4>
>
> **Reproducibility concerns.** We would like to stress that the complete implementation of PLIA$_t$, including all operations discussed in Sections 3 and 4, are provided in the supplementary material of our submission. This code also contains our complete experimental setup and automatically installs all necessary dependencies in an easy-to-run script as documented in the provided README file. PLIA$_t$ is a hyperparameter-free framework for inference tasks (cf. Section 5.1). For the neurosymbolic learning tasks (cf. Section 5.2) we provide all our optimal parameters in Appendix E. Upon acceptance, we will also make this implementation publicly available. In this regard we believe we follow best practices concerning reproducibility.
>
> We hope our answers have addressed your concerns and allow for an increase in score. Please let us know if there are any further concerns and we will gladly try to address them.

---

### Official Review · Reviewer_Ltsw · 2024-07-12

**Soundness:** 2
**Presentation:** 3
**Contribution:** 3
**Rating:** 5
**Confidence:** 2

**Summary:**

The paper addresses key challenges in applying neurosymbolic AI techniques, specifically in integer arithmetic. Leveraging the power of tensor operations and the fast Fourier transform (FFT), the authors propose a novel approach to perform probabilistic inference on integer-valued random variables. Central to their method is the tensor representation of the distributions of bounded integer-valued random variables, enabling efficient computation of distribution sums using FFT, thus significantly reducing computational complexity from quadratic to logarithmic time complexity. The study validates the effectiveness of this approach through experiments showing substantial improvements in inference and learning times, pushing the state of the art by several orders of magnitude. This contribution not only enhances computational efficiency but also facilitates gradient-based learning, which was previously challenging due to the discrete nature of integers.

**Strengths:**

1. The paper introduces a creative approach to tackling the computational challenges in probabilistic inference for integer arithmetic. The innovative use of tensor operations and the fast Fourier transform (FFT) to represent and compute the distribution of integer-valued random variables is a notable advancement. This method is original in its application of FFT in the log-domain for probabilities, which is a sophisticated adaptation not extensively explored in previous studies within the neurosymbolic AI domain.

2. The paper is exceptionally well-written, with a clear structure and logical flow that enhances readability. Technical concepts and methods are explained with precision, making the complex content accessible to readers who may not be intimately familiar with the technical details of probabilistic inference or FFT. Diagrams and examples further aid in understanding and illustrating key points effectively.

3. The significance of this work is manifold. Firstly, it addresses a critical bottleneck in neurosymbolic AI, opening up new possibilities for applying these techniques to more complex, real-world problems. The ability to perform differentiable operations on discrete data structures could have profound implications for the field, potentially influencing future research directions and applications.

**Weaknesses:**

1.While the paper briefly mentions the inherent #P-hardness of probabilistic inference, there is scant elaboration on how this impacts the scalability or applicability of the proposed method in practical, large-scale scenarios. A more detailed discussion of the practical limitations, potential computational bottlenecks, or scenarios where the method might not perform as expected would be invaluable for readers and practitioners looking to apply these techniques.
2. The clarity and depth provided in the theoretical formulation and experimental setup are excellent. However, the paper could be enhanced by including more detailed information on the implementation, such as the specifics of the software environment, libraries used, and parameter settings. This addition would aid in reproducibility and allow other researchers to more easily validate and build upon the work.
3. The paper could further detail the algorithmic complexity of the proposed method beyond the FFT application. While it highlights the reduction in time complexity, there's minimal discussion about space complexity or the computational overhead introduced by tensor manipulations and log-domain calculations.

**Questions:**

1.  Could the authors provide additional details about the computational environment, versions of libraries used, and precise parameter settings?
2.  How do the authors address potential underflow or overflow problems during computations, and what impact might these issues have on the method's accuracy and stability?
3. Are there plans or ongoing work to test the applicability of this approach in other fields that heavily rely on integer arithmetic, such as cryptographic algorithms or complex financial computations? Insights into such applications could significantly enhance the paper's impact.

**Limitations:**

One potential limitation is the paper's reliance on specific assumptions for the efficiency of FFT in the log-domain. While these assumptions are necessary for the mathematical framework, there is limited discussion on how sensitive the results are to deviations from these assumptions. Exploring the robustness of the proposed method under less ideal conditions or discussing potential modifications that could accommodate more general cases would strengthen the paper.

---

> ### Author Rebuttal · Authors · 2024-08-06
>
> We thank the reviewer for their thorough review and appreciate the raised concerns. We will start below by answering the main questions:
>
> 1. Our precise parameter settings are detailed in Appendix E for the neurosymbolic experiments (cf. Section 5.2), being learning rate, number of training epochs and neural architecture. As PLIA$_t$ itself is a hyperparameter-free framework, there are no other parameters to set when just performing inference in PLIA$_t$ (cf. Section 5.1).
> All other implementation details are provided in the supplementary code material (submitted together with the paper), where the precise computational environment, including the exact versions of all necessary libraries, are given. Moreover, our complete implementation of PLIA$_t$, its experiments and the precise parameter settings can also be found in the supplementary code material for ease of reproducibility. Finally, we will publicly release PLIA$_t$ upon acceptance.
>
>
> 2. The issue of numerical overflow or underflow is addressed in Section 2.2. There, we introduce a new variant of the well-known log-sum-exp trick for FFT computations in the log domain, called the fast log-conv-exp trick. Using the fast log-conv-exp trick PLIA$_t$ runs without any numerical stability issues.
>
>
> 3. We are currently looking into many different application domains for PLIA$_t$ as the scalability results from our experiments, where we observe orders of magnitude better performance, seem to indicate it is an enabling technology. We did not yet consider applications in cryptography or finance, but these are indeed promising directions! We thank the reviewer for bringing these to our attention.
>
>
> As for the weaknesses, we would like to address your raised concerns about limitations, complexity and computational overhead.
>
> **Limitations.** We acknowledge the discussion on the practical applicability of PLIA$_t$ could be extended. We propose to add the following discussion to the camera-ready version of the paper.
>
> The inherent #P-hardness of probabilistic inference does remain the main bottleneck for PLIA$_t$ and can be observed in our experiments. During the comparison between PLIA$_t$ and the state-of-the-art approach of Cao et al. [1] in Figure 5, we can see that PLIA$_t$ also starts to take a significant amount of time when the domain of the probabilistic integers grows. Hence, if the explicit computation of probabilistic integers with a very large domain is required, then PLIA can still struggle. However, PLIA$_t$ does scale orders of magnitude further (5 to be precise) than what was possible so far and considerably pushes the envelope on the state of the art.
>
> **Space complexity.** It is well-known that the FFT has, apart from the $N \log N$ runtime complexity, also $N \log N$ theoretical space complexity, again in contrast to a $N^2$ space complexity of the naive approach. Moreover, we performed an additional empirical comparison between Dice [1] and PLIA$_t$ with respect to memory allocation (see attached PDF in global rebuttal), where we indeed confirmed the better scaling behaviour of PLIA$_t$. Hence, PLIA$_t$ not only outshines Dice (the current state of the art) in terms of runtime but also in terms of memory usage. We will add this comparison to the appendix and mention it in the main paper in Section 5.1.
>
>
> **Computational overhead.** Finally, the computational overhead, including construction of the computational graphs for PLIA$_t$, is already included in the time measurements of all our experiments. For smaller domains, it is indeed possible that PLIA$_t$ on CPU or even Dice can outperform PLIA$_t$ on the GPU (Figure 5, bitwidth strictly below 5). However, it is clear that this computational overhead has a diminishing effect, as PLIA$_t$ clearly scales better than Dice for larger, more practical domain sizes (Figure 5, bitwidth above 5). Moreover, much of this overhead can be avoided by running PLIA$_t$ on the CPU for smaller domain sizes, where it performs on par or better than Dice (Figure 5, bitwidth strictly below 5). We will also make this point clearer in our experimental discussion in Section 5.1.
>
> [1] Cao, W. X., Garg, P., Tjoa, R., Holtzen, S., Millstein, T., & Van den Broeck, G. (2023). Scaling integer arithmetic in probabilistic programs. In Uncertainty in Artificial Intelligence (pp. 260-270). PMLR.

---

> > ### Comment · Reviewer_Ltsw · 2024-08-13
> >
> > Thank you for your response. I have decided to keep my score.

---

### Official Review · Reviewer_JM4B · 2024-07-13

**Soundness:** 4
**Presentation:** 3
**Contribution:** 3
**Rating:** 7
**Confidence:** 4

**Summary:**

This paper presents an approach to compute probability distributions over sums of random variables. To achieve this, the authors replace as slow quadratic computation with a Fourier transformation and a Hadamard product that can be computed in log-linear time.
The authors introduce the log-sum-exp trick to improve the stability of the computations. In this process, the authors note that their approach is differentiable, which allows them to connect with work on neurosymbolic AI.
The authors present metrics that can be computed, and finally the authors present an empirical evaluation which demonstrates the accuracy and speed improvements of their approach.

**Strengths:**

The connection to neurosymbolic AI is very interesting and in my opinion the most important part of the paper. The performance of the method both in accuracy and speed are significant. The implementation using GPUs and DNN libraries are valuable for the community.

**Weaknesses:**

My only concern with this paper, is that their work shares many similarities with the work in [1]. The authors do not cite this paper, which includes the convolutions and Fourier transform motivation which is known, as well as the use of the log-sum-exp trick. Providing a citation and a more thorough differentiation would be extremely valuable and maybe even required.

[1] Parker, M. R., Cowen, L. L., Cao, J., & Elliott, L. T. (2023). Computational efficiency and precision for replicated-count and batch-marked hidden population models. Journal of Agricultural, Biological and Environmental Statistics, 28(1), 43-58.

**Questions:**

Do you do a special treatment for the case where the pmf is zero i.e., where the log is -inf? or do you simply allow it and let the exponential handle that? A small paragraph on that would be interesting.

**Limitations:**

no issues

---

> ### Author Rebuttal · Authors · 2024-08-06
>
> Thank you for your review and for bringing the work of [1] to our attention!
>
> Firstly, we would like to clarify that we do not introduce the log-sum-exp trick as this is indeed a well-known trick to avoid numerical instabilities when summing probabilities in log space. Instead, we introduce a similar trick, dubbed the fast log-conv-exp trick, that avoids numerical instabilities when applying the FFT on probabilities in log space.
>
> Second, we would like to stress the differences between PLIA$_t$ and [1]. Both utilise the FFT to improve computational efficiency of probabilistic inference.
>
> 1. PLIA$_t$ and the method of [1] use the FFT for probabilistic inference on different applications. PLIA focuses on scaling probabilistic inference in linear integer arithmetic, while [1] uses the FFT for N-mixture models and applies it to the open-population model of [2].
>
> 2. While [1] does discuss numerical instability of computing convolutions in log space, their proposed solution differs from our proposed fast log-conv-exp trick in the following way. As can be seen from algorithm 2 in [1], their approach is to use the traditional log-sum-exp trick inside each iteration of the FFT and inverse FFT, which prevents using any off-the-shelf FFT algorithm and adds computational overhead (note that FFT algorithms are extremely tough to implement efficiently). In contrast, our trick exploits the linearity of the Fourier transform to show it is possible to simply rescale the in- and outgoing log probabilities by their maximum to prevent numerical instabilities. Just like the linearity of the summation does for the log-sum-exp trick. This means we go to and from linear space only once instead of at every iteration in the FFT. Our approach has the advantage of being implementation-agnostic, allowing for off-the-shelf and highly optimised FFT implementations to be used.
>
>
> 3. The proposed tensorised representations and operations of PLIA$_t$ are crucial for the measured increase in performance. Additionally, they also lead to out-of-the-box differentiability. Both of these aspects are not discussed nor examined by [1]. Moreover, [1] only reports an improvement of 6x-30x in computation time for two specific population models, while we observed improvements in the order of 100 000x in our benchmarks compared to the state of the art.
>
>
> 4. The work of [1] only proposes a solution for computing numerically stable convolutions (discussed in points 2 and 3 above). We do provide further operations that can be performed efficiently, e.g. multiplication by scalars and probabilistic branching. This is not discussed at all in [1].
>
> We will add a reference to [1] to the paper and also include the above discussion.
>
> Third, to answer your question, our implementation does allow for -infinity in case the PMF is zero. We utilise the numpy implementation np.inf that adheres to the IEEE 754 [3] industry standard for representing infinity as a floating-point number. Moreover, this implementation is compatible with all operations of our deep learning backend, TensorFlow.
>
> We hope to have clearly differentiated PLIA from the cited work and to have addressed all your concerns. If so, we hope you are open to the idea of raising your score. If not, we will gladly try to discuss any remaining concerns.
>
> [1]  Parker, M. R., Cowen, L. L., Cao, J., & Elliott, L. T. (2023). Computational efficiency and precision for replicated-count and batch-marked hidden population models. Journal of Agricultural, Biological and Environmental Statistics, 28(1), 43-58.
>
> [2] Dail, D., Madsen, L. (2011). Models for estimating abundance from repeated counts of an open metapopulation. Biometrics 67(2):577–587.
>
> [3] IEEE Standard for Binary Floating-Point Arithmetic. 1985.

---

> ### Comment · Reviewer_JM4B · 2024-08-12
>
> Thanks for the reviewer for the detailed response. I acknowledge I have read the rebuttal and I have raised my score accordingly.

---

### Author Rebuttal · Authors · 2024-08-07

We thank the reviewers for their valuable comments and suggestions to improve the paper. Our rebuttal to the individual reviews can be found directly under each review. We also attached a plot (as pdf) used to address Reviewer Ltsw's concern regarding memory usage.We discuss this issue just below.

We hope the rebuttal clears up some of the concerns and allows the reviewers to reconsider their inital scores. Please let us know if there are remaining imprecisions/unclarities.


#### **Space Complexity**

We thank Reviewer Ltsw for raising the concern of space complexity and memory usage, and acknowledge this aspect was not properly covered. While it is well-known that the space complexity of the FFT is also $N \log N$, just as its runtime, an empirical comparison was still missing. To this end, we analysed the memory consumption of PLIA$_t$ for the inference tasks in Section 5.1 and compared it to the state of the art, Dice [1]. The result can be seen in the attached PDF showing that PLIA$_t$ indeed scales better and seems more efficient in terms of memory usage compared to the state of the art across all inference benchmarks. We will add the attached figure to the appendix and refer to it during our experimental discussion in Section 5.1.


[1] Cao, W. X., Garg, P., Tjoa, R., Holtzen, S., Millstein, T., & Van den Broeck, G. (2023). Scaling integer arithmetic in probabilistic programs. In Uncertainty in Artificial Intelligence (pp. 260-270). PMLR.

---

### Decision · Program_Chairs · 2024-09-25

**Decision:**

Accept (poster)

**Comment:**

This paper describes a simple idea for probabilistic inference over linear, integer systems, based on the well-known property that the distribution of a linear combination of random variables can be performed using the Fourier representation. This allows the authors to frame their inference process using tensor operations, with a useful byproduct being that the entire computation becomes differentiable and usable in learning. Empirically, they show very high speedups and show the application of their ideas to neurosymbolic programming.
The ideas in the paper are straightforward, with the result that some of the text feels obvious — the discussion of zero-padding to match vector lengths, for example, and even the authors’ log-sum-exp trick contribution. However, in their application they manage to improve over very recent work in significant ways.  One reviewer pointed out a closely related publication, which the authors’ response indicated they would incorporate.  Otherwise, reviewers were agreed on the potential impact of the work, with a few concerns that were generally well addressed by the author response.